# Preliminary Analysis on the Hydrostatic Stability of a Self-Aligning Floating Offshore Wind Turbine

**Diane Scicluna [1], Claire De Marco Muscat-Fenech [1], Tonio Sant [1,*], Giuliano Vernengo [2] and Tahsin Tezdogan [3]**

1   Department of Mechanical Engineering, University of Malta, MSD 2080 Msida, Malta
2   Department of Electric, Electronic, Telecommunication Engineering and Naval Architecture (DITEN), Polytechnic School, University of Genoa, 16145 Genoa, Italy
3   Department of Naval Architecture, Ocean and Marine Engineering, University of Strathclyde, Glasgow G1 1XQ, UK
*   Correspondence: tonio.sant@um.edu.mt; Tel.: +356-2340-2437

**Abstract:** There exist vast areas of offshore wind resources with water depths greater than 100 m that require floating structures. This paper provides a detailed analysis on the hydrostatic stability characteristics of a novel floating wind turbine concept. The preliminary design supports an 8 MW horizontal-axis wind turbine with a custom self-aligning single-point mooring (SPM) floater, which is to be constructed within the existing shipyard facilities in the Maltese Islands, located in the Central Mediterranean Sea. The theoretical hydrostatic stability calculations used to find the parameters to create the model are validated using SESAM®. The hydrostatic stability analysis is carried out for different ballast capacities whilst also considering the maximum axial thrust induced by the rotor during operation. The results show that the entire floating structure exhibits hydrostatic stability characteristics for both the heeling and pitching axes that comply with the requirements set by the DNV ST-0119 standard. Numerical simulations using partial ballast are also presented.

**Keywords:** FOWT; self-alignment; SPM; renewable energy; wind energy





## 1. Introduction

The increasing demand for energy and the increasingly evident consequences of climate change are fuelling significant interest in alternative energy sources such as wind energy. Wind energy has been used for centuries through the use of windmills, which were further developed into onshore wind turbines to generate electricity. Technological developments have seen the rapid growth of the wind energy industry from onshore structures installed on land to offshore structures, including both bottom-fixed and floating structures [1].

Offshore wind turbines provide significant benefits when compared to onshore systems. Although onshore structures are cheaper and present less technological challenges to install and maintain, such installations provide a limitation with respect to the large amount of space required, a limitation that is not present for offshore turbines due to the vast seas and oceans found all over the world. Offshore structures also create less visual and noise pollution than onshore, resulting in higher levels of public acceptance associated with the deployment of offshore wind energy structures. Furthermore, with increasing distance from the shore, faster, stronger and more consistent wind speeds are present, allowing for increased opportunities for steadier power generation [2].

The fast growing interest in wind energy has resulted in a larger demand for upscaled turbines, with sizes expected to reach up to 15–20 MW, as highlighted in [3]. The increased turbine sizes require larger platforms and substructures, which are difficult to incorporate in onshore structures [4].

A number of studies have analysed the challenges related to the technical, economic and design characteristics that are associated with the design of larger wind turbines [3–6]. The main issue with upscaling is the increase in rotor mass, which may be reduced using alternative lightweight components and novel design concepts [7].

Several support systems currently exist in the offshore wind turbine (OWT) industry. These can be classified into bottom-fixed structures and floating structures. Bottom-fixed structures are connected directly to the seabed, a feature which imposes a feasible limit to the sea depth in which the structures can be installed. Generally, bottom-fixed structures are restricted to a sea depth of approximately 50 m as deeper waters lead to an exponential increase in costs [2]. Floating offshore wind turbine (FOWT) structures use mooring lines and anchors to connect to the seabed, allowing them to be installed in deeper waters with lower costs than bottom-fixed structures [8].

Both types of offshore wind energy platforms are similar to structures used in the offshore oil and gas industry, with modifications included to accommodate the wind turbine [9]. The three main types of floating offshore platforms are the semi-submersible, the tension leg platform (TLP) and the spar-buoy as shown in Figure 1 [2].

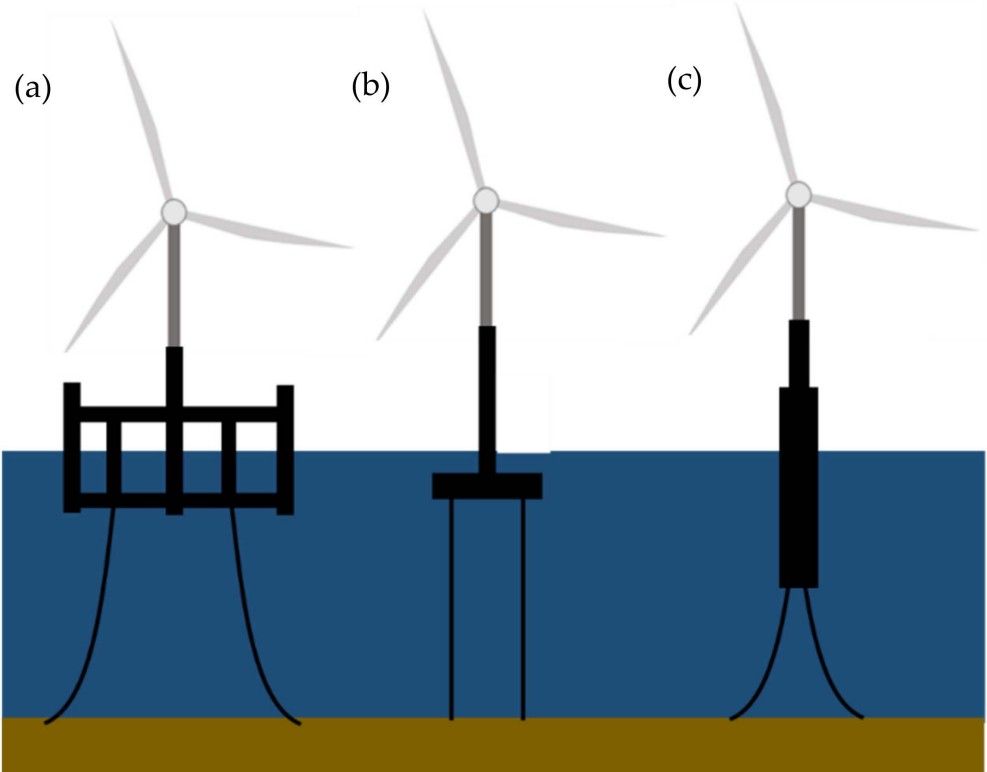

**Figure 1.** (**a**) Semi-submersible, (**b**) TLP (**c**) Spar-buoy.

It is vital for FOWT structures to reduce the degree of motion and achieve stability given the harsh environmental conditions present at sea. Stability is achieved using three main mechanisms: (1) ballast-stabilisation, (2) buoyancy-stabilisation and (3) mooring-stabilisation [10]. Table 1 shows the different FOWT types with the stabilising techniques used [6,7]. Although it is often stated that each structure uses only one method for stabilisation, in reality all floating platforms use a combination of the aforementioned stabilisation techniques in order to further enhance their stability characteristics [11].

**Table 1.** FOWT stabilising mechanisms.

| FOWT | Description and Stabilising Mechanism |
|---|---|
| Spar-buoy | A long cylindrical structure with a low WPA, which is ballast stabilised. |
| TLP | A central column connected to a submerged buoyant platform, which is mooring stabilised. |
| Semi-submersible | A number of columns connected with braces, pontoons or platforms that are buoyancy stabilised. |

In the offshore wind industry, the definition of self-alignment in floating structures is the ability of a floating platform to weathervane with the prevailing wind and wave direction without requiring an active yaw system. In general, a yaw mechanism is used in wind turbines to align the rotor plane to continuously face the direction of the wind flow to increase the generation of power. Misalignment between the wind flow direction and the rotor results in decreased efficiency. A self-aligning FOWT structure no longer requires a complex yaw mechanism.

By incorporating self-alignment characteristics during the preliminary design stage, significant cost reductions can be obtained [12]. In the current industry, mooring technology is being used to include self-aligning FOWT structures. Mooring costs account for a considerable part of the overall costs of floating offshore platforms; therefore, the type of mooring systems used is an important factor in FOWT design [13].

The most common type of mooring system is known as the multi-point mooring (MPM) system. It makes use of multiple mooring lines distributed geometrically around the floating structure. The MPM systems fix the structure in position and eliminate the possibility of weathervaning. Single-point mooring (SPM) systems are now being introduced in the FOWT industry. In the case of SPM systems, the mooring system is connected to the floating structure at only a single point. The SPM systems use either a single mooring line or multiple mooring lines connected at the seabed that come from a single point placed on the floating platform [14]. The aforementioned mooring systems are shown schematically on a semi-submersible FOWT structure in Figure 2.

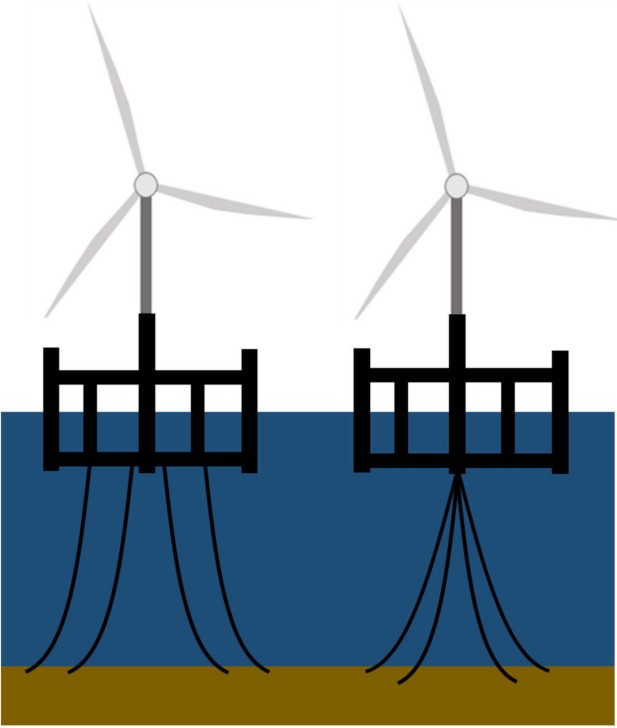

**Figure 2.** Schematic diagram of the MPM and SPM systems.

Initially, SPM systems were known as passive weathervaning mooring systems as such systems allow the floating structure to self-align with the prevailing wind and wave conditions. Weathervaning reduces wear and damage on the mooring lines and minimises the need for tug assistance as the structures are not exposed to large hydrodynamic and aerodynamic forces [14].

Although the self-alignment properties present significant benefits to FOWT structures, a small degree of misalignment between the direction of wave propagation and direction of wind flow will always be present. Such a misalignment leads the turbine to operate with a yawing error. This may lead to loss in the rotor aerodynamic efficiency if the error is considerably large.

Although SPM systems increase the self-alignment characteristics in FOWTs, additional features still need to be included into the design to supplement the weathervaning characteristics of the structure and improve the self-alignment response behaviour of the wind turbine to the prevailing wind condition.

Single-point mooring systems are increasing in popularity, an example of which is the HyStOH concept [15]. This concept uses an SPM system for self-alignment but also includes a novel tower design shaped like an aerofoil. The tower design provides the necessary forces for self-alignment under the prevailing wind conditions. Despite the positive results obtained from previous research projects, the prevailing conclusion was that the self-alignment characteristics of the structure were limited by current velocities and yaw errors. Therefore, it was concluded that including additional techniques to increase the self-alignment moments were necessary [9,11].

Other designs that use SPM systems are Eolink and X1 Wind. Eolink consists of a four-legged space frame shaped as a pyramid and having a square base to reduce the weight of the structure. Similarly, X1 Wind makes use of a structure shaped as a triangle to support the turbine, instead of the conventional tower structure, combined with a customised SPM system called PivotBuoy. Although emerging designs are already incorporating self-alignment characteristics and SPM systems, such designs present construction challenges for existing shipyard facilities available on small islands, such as those found in the Maltese Islands, as will be further discussed in Section 3.

The scope of this paper is to: (1) present a brief overview of floating offshore wind turbines (FOWTs) and introduce self-aligning turbines using single-point mooring (SPM) technology and (2) to assess the large angle hydrostatic stability characteristics of a novel self-aligning 8 MW floating wind turbine having a dual-hull configuration with dynamic ballasting and which can be constructed in existing shipyards having a limited width and draught.

## 2. Design Standards for Floating Turbines

The FOWT industry is rapidly developing, with a significant number of installations and emerging projects. In order to keep up with the fast FOWT development, a need has emerged for more detailed standards to be drafted to ensure safer, successful and long-term operations.

The International Maritime Organisation (IMO) presents the IMO MSC 267(85), known as the parent of all standards produced by regulation organisations worldwide. The IMO MSC 267(85) presents the intact stability criteria for various structure types [16].

The stability of an offshore structure is found using the righting moment and heeling moment curves, taken about the critical axis with free surface effects also taken into consideration [16].

Parameters such as the minimum and maximum righting lever (GZ) area and GZ values, initial metacentric height (GM), and wind and rolling criteria are used to identify whether a structure is safe and ready for deployment [16].

The DNV standards referred to as Intact OS-C301 and ST-0119, based on the IMO standard, are commonly used for FOWT structures.

The main requirements for floating offshore structures can be seen in Table 2.

**Table 2.** Requirements set by standards.

| | IMO MSC267(85) MODU [13] | DNV OS-C301 Sect. 4.3 [17] | DNV ST-0119 Sect. 10.2.3 [15] |
|---|---|---|---|
| For column stabilised units | ✓ | ✓ | ✓ |
| For semi-submersibles | - | - | ✓ |
| Area under righting moment curve to the second intercept or downflooding angle in excess of the area under the wind heeling moment curve | | | |
| | ≥30% | ≥40% | ≥130% |
| Righting moment curve should be positive over range of angles from upright to the second intercept. | | | |
| | ✓ | ✓ | ✓ |

Therefore, given that the model considered is a column-stabilised semi-submersible unit, DNV ST-0119 is used to ensure that the required standards are met. The righting moment and heeling moment curves developed from [18] as defined by the DNV ST-0119 standard are shown in Figure 3.

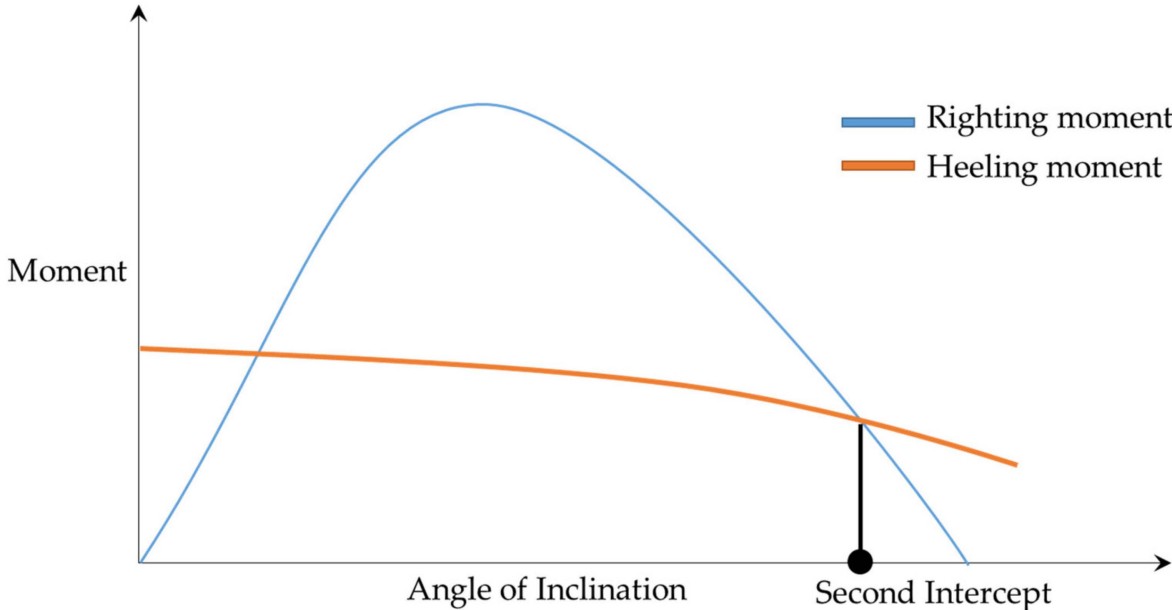

**Figure 3.** Righting and heeling moment curves.

The righting moment depends greatly on the restoring capabilities of the structure. The wind heeling moment is generated by the wind acting on the structure at a distance from the centre of rotation.

The wind loads on the rotor can be calculated using Equation (1):

$$F_T = 0.5\rho C_T A_{rotor} U^2 \tag{1}$$

where

$F_T$ is the thrust force, N
$\rho$ is the density of air, kg/m$^3$
$C_T$ is the thrust coefficient of the rotor
$A_{rotor}$ is the swept area of the rotor, m$^2$
$U$ is the rated wind speed of the turbine, m/s

The force of the wind acting on the structure can be calculated using Equation (2) from DNVGL-0S-C301 [17]:

$$F_S = 0.5\rho C_S C_H A_P U^2 \tag{2}$$

where

$F_S$ is the wind load on the structure, N

$\rho$ is the density of air, kg/m$^3$

$C_S$ is the shape coefficient depending on the shape of the structural member exposed to the wind

$C_H$ is the height coefficient depending on the height above sea level of the structural member exposed to the wind

$A_P$ is the projected area of all exposed surfaces in either the upright or heeled condition, m$^2$

$U$ is the wind speed, m/s

The wind heeling moment can then be calculated using Equation (3):

$$M_{wind} = F_T \times (z_{hub} - z_{COB}) \cos^2(\theta) \tag{3}$$

where

$M_{wind}$ is the wind heeling moment, Nm

$z_{hub}$ is the vertical distance from the base of the tower plus the freeboard of the structure, m

$\theta$ is the pitch angle, °

## 3. Proposed Design

Malta's options for renewable energy are greatly limited by its size and geographical locations. Currently, the only possible options are offshore wind energy, solar photovoltaic sources, solar thermal energy and energy from waste. One of the most important issues associated with introducing offshore wind energy around the Maltese Islands, located in the Central Mediterranean Sea, is the lack of shallow waters nearshore. The deep nearshore waters (>100 m) coupled with the significant lack of available space present in Malta eliminate both onshore and bottom-fixed structures as viable options. Therefore, the restrictions demand the use of FOWTs to be able to exploit the offshore wind resources available on a large scale.

To create the optimised FOWT design for Malta, the limitations present in the Maltese Archipelago must be acknowledged. The spatial limitations and deep nearshore waters impose restrictions both on the size of the structure and the type of FOWT possible. Furthermore, given that FOWTs require large support structures and regular maintenance, the dependability of port infrastructure as well as the possibility of developing new ports to support the emerging FOWT industry merit considerable attention.

The current designs in the FOWT industry present logistical difficulties for Malta's ports, primarily due to the large width of the floaters and particularly in the case of semi-submersible floaters currently being tested for supporting megawatt-scale wind turbines (greater than 6 MW).

The proposed concept incorporates novel design characteristics for both the improvement of the self-alignment capabilities of the structure and the removal of a rotor-yaw mechanism. Furthermore, the proposed design features a decreased width compared to the current floater designs found in the industry, allowing it to be manufactured in local ship-building docks. Investigating the effect of mooring is not within the scope of this paper; however, the structure is also intended to incorporate single-point mooring (SPM) technology as an additional self-alignment characteristic.

The proposed model is shown in Figure 4.

The features of the proposed design can be summarized as follows:

- Tower: Conventional tower replaced by a lighter space frame made up of beams.
- Rotor-Nacelle Assembly (RNA): 8 MW downwind turbine without the need for a costly yaw mechanism.
- Columns: Aero-hydrodynamic vertical columns shaped like aerofoils for self-alignment with wind.
- Dual-hull floating platform, which is radically different than other floating turbine platform concepts under development: Narrow hulls are included to enable construction in existing shipyard docks that are too narrow for other floating turbine platforms.

Apart from also reducing the draught requirements, the hulls enable improved self-aligning capabilities with waves. A dynamic ballasting system is also integrated.

- Deck space for integrated energy storage if required and additional infrastructure.

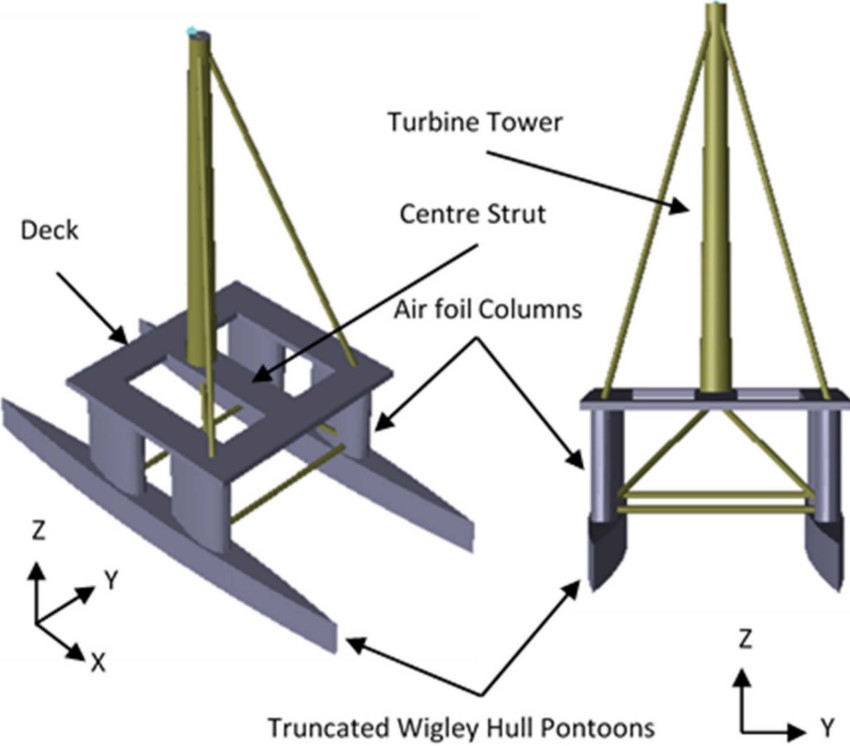

**Figure 4.** Proposed Model.

The dimensions and geometry of the structure were found using an in-depth parametric study. The fundamental fixed parameters, such as the maximum possible width and the draught of the structure, were defined before carrying out the parametric study. The parameters were set to remain within the limits of the intended on-site construction ship-building dock found within the Maltese Islands. The maximum length, width and draught of the ship-building dock were 362 m, 62 m and 9.14 m, respectively. Additional limitations as specified within DNV standards were also taken into consideration and a PASS/FAIL criteria was set up to immediately point out any design issues. The parameters of the structure can be seen in Table 3.

**Table 3.** Model Parameters.

| Parameter | Value |
|---|---|
| Power rating, rated wind speed | 8 MW, 12.5 m/s [19] |
| Rotor orientation, configuration, diameter | Downwind, 3 blades, 164 m |
| Tower height, base diameter, top diameter | 92.5 m, 7.7 m, 5 m |
| Rotor diameter, Nacelle diameter | 4 m, 7.5 m |
| Pontoon: length, width, height | 121 m, 8 m, 9.9 m |
| Column: height, number of columns | 25 m, 4 |
| Deck: height, length, breadth | 2 m, 55 m, 60 m |
| Total mass (no ballast), displaced volume (no ballast) | 5254 t, 5125.5 m$^3$ |
| Draught | 4 m |
| COG (w.r.t free water surface) | (1.6 m, 0 m, 23.3 m) |

After finalising the parameters of the model, the structure was created using a combination of plates and beams in DNV SESAM® GeniE 64 V8.0-21 [20]. The compartments required for the ballast tanks were also created. The model was then meshed accordingly for hydrostatic analysis in SESAM® HydroD V4.10-01 [20].

## 4. Analysis Layout

The final numerical model was designed in SESAM® GeniE [20], and the hydrostatic stability was analysed in SESAM® HydroD [20]. The four different analyses carried out are shown in Table 4.

**Table 4.** Analysis Layout.

| Stability Analysis | Condition | Abbrev. | Ballast | Standard |
| --- | --- | --- | --- | --- |
| Upright Stability | Validation Process (VP) | SA: VP | | |
| Large Angle Stability | No Wind (NW) | SA: NW | 0–100% | DNV-ST-0119 Sect. 10.2.3 |
| Large Angle Stability | Rated Wind (RW) | SA: RW | 0–100% | DNV-ST-0119 Sect. 10.2.3 |
| Large Angle Stability | Partial Ballast (PB) | SA: PB | 0–100% | DNV-ST-0119 Sect. 10.2.3 |

The validation compared the theoretical calculations carried out using Excel 2013® with values obtained from GeniE® and HydroD®. The validation (SA: VP) analysis was under lightship conditions and on an even keel; therefore, any initial trim angle present in the structure was neglected. The validation process compared the results obtained from the models implemented theoretically in Excel® and numerically in SESAM®.

The scope of the SA: NW was to assess the large angle hydrostatic stability of the structure under no wind conditions. SA: NW was carried out in HydroD® where the model was analysed under different loading conditions with ballast capacities ranging from lightship at 0% capacity to full and down capacity at 100% when the turbine was not in operation.

SA: RW was used to analyse the large angle stability of the structure to verify that the structure met all safety requirements set by DNV ST-0119 [18] under the effect of the wind loads acting on the structure and the rotor thrust in the pitching and the heeling axes. SA: RW investigated the effect of wind moments on the proposed design at a rated wind speed of 12.5 m/s. The wind heeling moment was considered along two directions as shown in Figure 5:

- The wind direction perpendicular to the rotor plane, to consider pitching.
- The wind direction parallel to the rotor plane, to consider heeling.

The ST-0119 standard specifies that the wind heeling moment needs to be considered only when the wind is acting perpendicular to the rotor plane.

The proposed design is a dual-hull structure and can be likened to a catamaran. Research carried out by Deakin, 2003 [21] shows that the majority of incidents where catamarans capsized were wind induced (pitchpoling) capsizing, i.e., the capsizing by pitching forward. Therefore, given that such structures are prone to capsizing by pitchpoling and that the proposed design is intended to weathervane, verifying that the structure is stable under frontal wind conditions in the pitching axis is considered important.

As the structure is intended to self-align, the whole structure will position itself to align with the prevailing wind and wave conditions. Therefore, realistically, there is no scenario where the proposed design will encounter winds in the parallel direction to the rotor plane while the turbine is in operation.

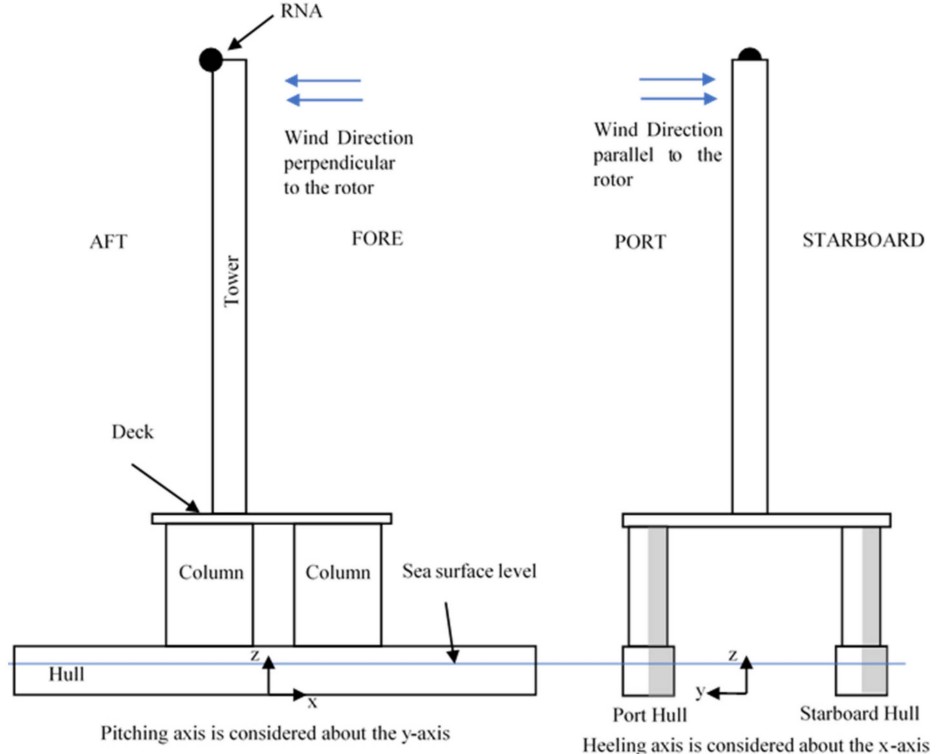

**Figure 5.** SA: RW wind direction.

Nonetheless, there is a possibility of the wind direction aligning with the rotor plane while the FOWT structure is being towed to position at sea. At this point, the structure will not be able to self-align and so may experience heeling. Therefore, the stability characteristics of the structure in the heeling axis under rated wind conditions was also considered.

From the output of SA: NW and SA: RW, a third analysis denoted as SA: PB was developed to evaluate how partial ballasting can reduce the initial pitch angle present in 80–100% ballast. The partial ballasting exercise was carried out by varying the level of ballast in the tanks located towards the aft of the structure, while maintaining the same level of ballast in the fore tanks in order to reduce the trim angle so that the permissible motion of the structure was not exceeded. The fore (green) and aft (blue) tanks are shown in Figure 6.

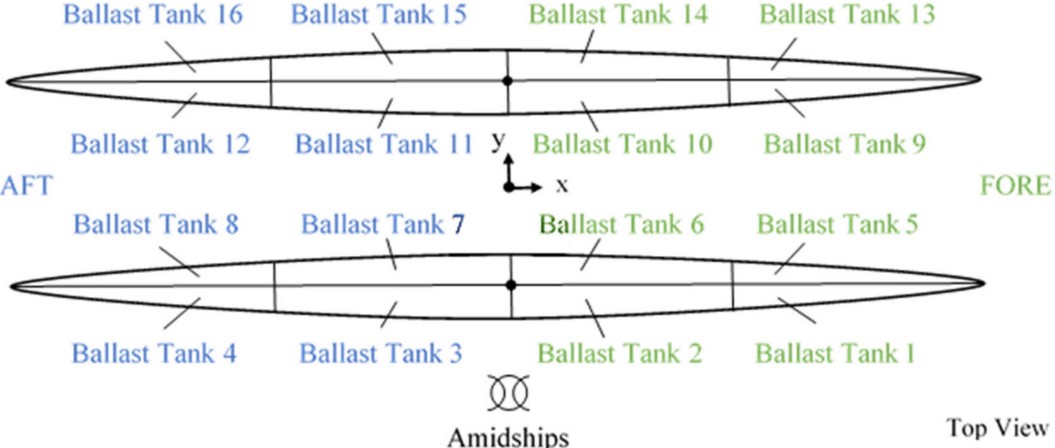

**Figure 6.** Ballast tank layout.

The analysis was carried out under the loading conditions specified in Table 5.

**Table 5.** SA: PB test conditions.

| Test Case | Fore Tank % Ballast | Aft Tank % Ballast |
|-----------|---------------------|--------------------|
| 1 | 80% | 85% |
| 2 | 80% | 90% |
| 3 | 80% | 95% |
| 4 | 80% | 100% |

## 5. Results and Discussion

### 5.1. SA: VP

The validation analysis for SA: VP used an initial upright hydrostatic analysis and compared the theoretical results with the results from the geometric modeller GeniE® and the hydrostatic analysis software HydroD®.

General naval architectural procedures usually consider the structure to be on an even keel for the validation of parameters, thereby neglecting any initial trim present while the structure floats in still water conditions [22]. Therefore, for validation, the model was analysed on an even keel under lightship conditions to ignore free surface effects (FSE). The validation results are presented in Table 6.

**Table 6.** Results of Validation Process, SA: VP.

| Parameters | Excel® Value | GeniE® Value | GeniE® %Difference | HydroD® Value | HydroD® %Difference |
|------------|-------------|--------------|--------------------|---------------|--------------------|
| $\Delta$ (t) | 5254 | 5268 | −0.3% | 5273 | −0.4% |
| $T$ (m) | 4.0 | * | * | 4.0 | 0 |
| $\nabla$ (m$^3$) | 5126 | * | * | 5144 | −0.4% |
| $LCG$ (m) | 1.6 | 1.5 | 0.4% | 1.5 | 1.7% |
| $TCG$ (m) | 0 | 0 | 0% | 0.0 | 0.0% |
| $VCG$ (m) | 33.2 | 33.2 | −0.9% | 32.7 | 0.6% |
| $LCB$ (m) | 0 | * | * | 0 | 0% |
| $TCB$ (m) | 0 | * | * | 0 | 0% |
| $VCB$ (m) | 2.0 | * | * | 2.1 | 4.9% |
| $GM_L$ (m) | 152 | * | * | 150 | 1.3% |
| $GM_T$ (m) | 127 | * | * | 120 | 5.4% |

* Not available by GeniE®.

As presented in Table 6, the values of mass, displaced volume and centre of gravity ($LCG$, $TCG$, $VCG$) obtained from the parametric analysis all compared well with the GeniE® and HydroD® values with a small percentage difference.

The minimal percentage difference was recorded between the parametric analysis and HydroD®. The maximum variation was noted for the transverse metacentric height. The difference was due to the limited number of strips considered for the integration of the second moment of area in the parametric study, thereby resulting in a 5.4% margin of error.

The results of the parametric analysis concluded that the theoretical calculations were sufficiently validated by the results obtained from the both the geometric modeller, GeniE®, and the hydrostatic stability software, HydroD®.

### 5.2. SA: NW

The hydrostatic values obtained from SA: NW can be seen in Table 7. The structure experienced a small initial trim angle of +0.5° towards the x-direction due to the weight of

the support beams connecting the tower and the deck. From 80% ballast to 100% ballast, the initial trim angle of the structure increased sharply from 0.5° to 15.2°, reaching 25.2° at 100% ballast.

**Table 7.** Hydrostatic values obtained from SA: NW.

| Ballast | $T$ (m) | $\Delta$ (t) | WPA (m$^2$) | $\Theta$ (°) | $GM_L$ (m) | $GM_T$ (m) |
|---|---|---|---|---|---|---|
| 0% | 3.3 | 5273 | 1539 | 0.5 | 185 | 150 |
| 10% | 4.2 | 6562 | 1539 | 0.5 | 136 | 121 |
| 20% | 5.0 | 7851 | 1539 | 0.5 | 114 | 101 |
| 30% | 5.8 | 9140 | 1539 | 0.5 | 99 | 87 |
| 40% | 6.6 | 10,428 | 1539 | 0.5 | 87 | 77 |
| 50% | 7.4 | 11,717 | 1539 | 0.5 | 78 | 69 |
| 60% | 8.2 | 13,006 | 1538 | 0.5 | 70 | 62 |
| 70% | 9.1 | 14,295 | 1538 | 0.5 | 64 | 57 |
| 80% | 14.6 | 15,584 | 884 | 15.2 | 17 | 28 |
| 90% | 17.6 | 16,873 | 693 | 18.9 | 19 | 21 |
| 100% | 22.7 | 18,162 | 497 | 25.2 | 14 | 13 |

Where $T$ = draught (m), $\Delta$ = mass (t), WPA = waterplane area (m$^2$), $\Theta$ = pitch angle (°), $GM_L$ = longitudinal metacentric height and $GM_T$ = transverse metacentric height.

It could be noted that at a ballast of 80% capacity, the draught exceeded the height of the hull/pontoons and the columns began to submerge. Under these conditions, a reduction in the waterplane area (WPA) occurred. The reduced WPA caused the Centre of Flotation (COF) of the structure to shift further aft of the tower position. This shift produced the large trim angles at the 80–100% ballast condition. Table 7 also shows that both transverse and longitudinal GM values decreased dramatically at 80% ballast, which was also attributed to the sudden decrease in WPA caused by the submerging of the pontoons.

The draught and metacentric heights from SA: NW shown in Table 7 are slightly higher than those obtained from SA: VP. The difference arose primarily due to the initial trim, which was accounted for in SESAM® but excluded from the parametric study carried out in SA: VP.

From the results of SA: NW, it could be concluded that the ballast compartments could not be filled equally past 70% due to the increase in pitch angle. To prevent the excess trim present between at 80%–100% ballast, dynamic balancing must be used. The individual ballast tanks must be filled accordingly to offset the large trim obtained.

The GZ curves for both the heeling and the pitching axes can be seen in Figures 7 and 8, respectively. From the GZ curves obtained under the ballast conditions of 0–40%, the traditional shape of a GZ curve was preserved as presented previously in Figure 2.

On the other hand, when the ballast was increased to between 50 and 100%, the shape of the GZ curves started to flatten, most prominently between 80 and 100% ballast. The unusual shapes of the graphs could be attributed to the increased pitch angle due to the significant reduction in WPA.

In the heeling axis, the graphs of 50–70% ballast were seen to peak sharply before decreasing. The GZ curves obtained for the 80–100% ballast showed smaller peaks around the same heel angles. The peaks were due to the different components of the structure such as the columns (A) and the deck being immersed (B, C) under water. The shape of the peaks shown in ballasts 80%–100% were not as sharp due to the larger pitch angles present. At ballast conditions from 0% to 50% ballast in the heeling axis, the range of positive stability increased with ballast. This was seen from the vanishing point, i.e., the point where the GZ curve crossed the horizontal axis, which occurred at a heel angle of 40° at lightship conditions to approximately 67° at 50% ballast. The larger the range of positive stability,

the better the structure's ability to withstand a capsizing moment. The range of positive stability started to decrease when the ballast capacity reached 60% up to 100%.

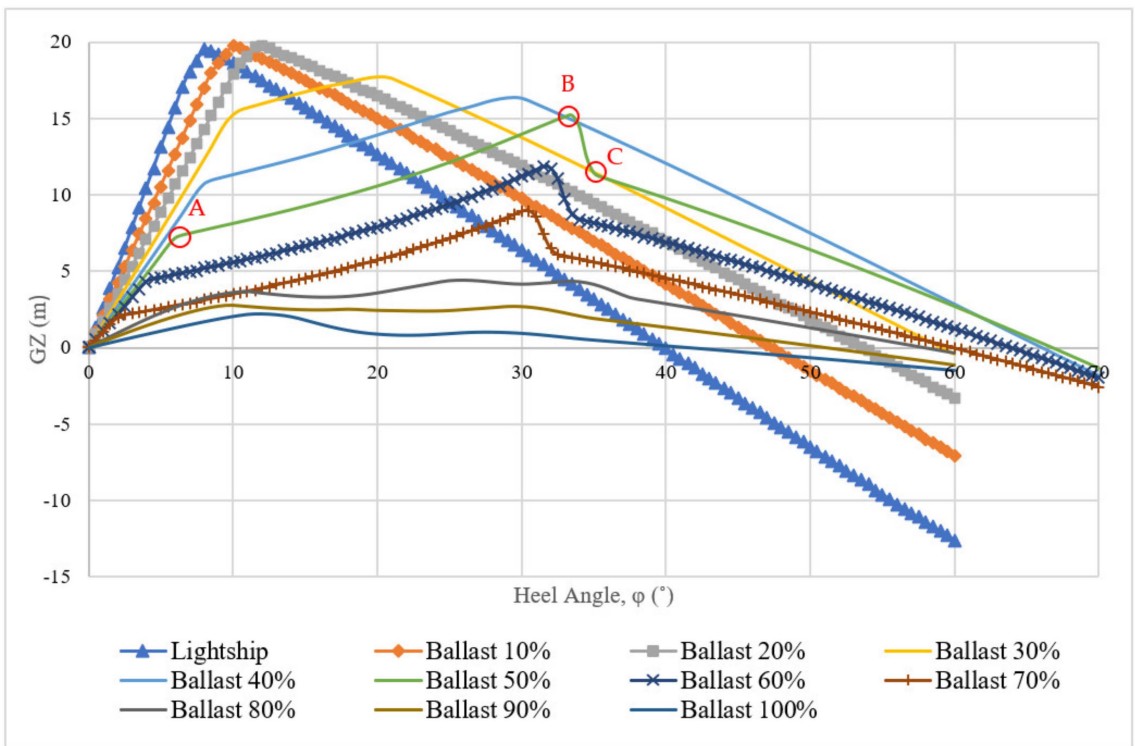

**Figure 7.** GZ curves for heeling where A refers to the submersion of the columns and B, C represent the submersion of the deck of the structure.

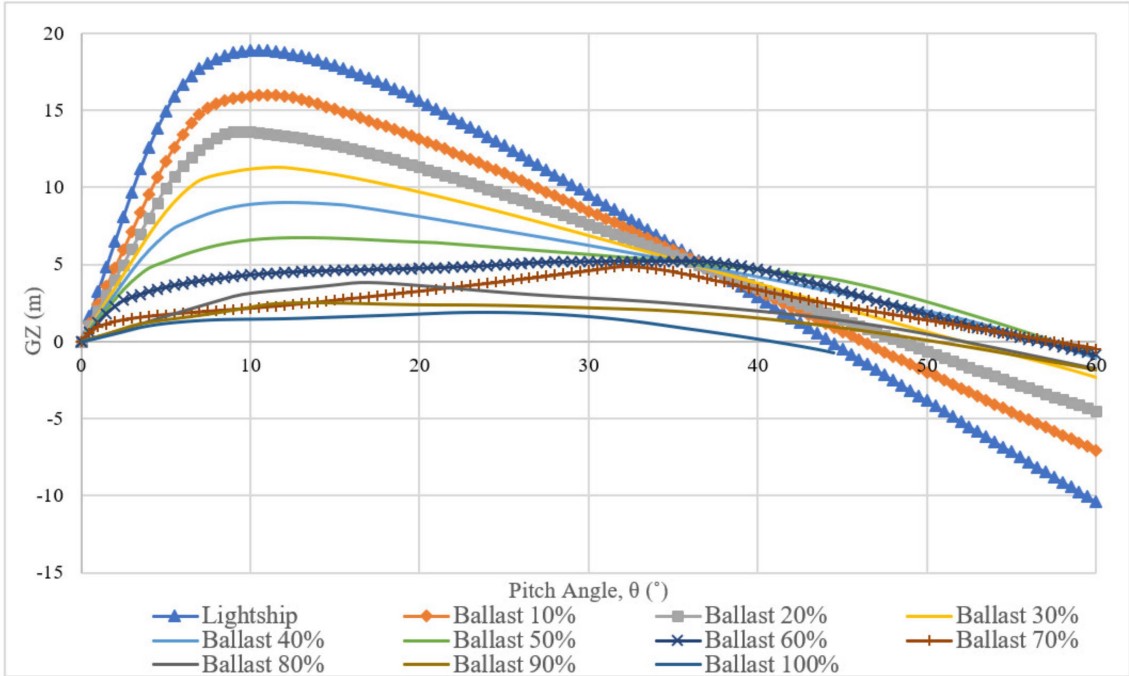

**Figure 8.** GZ curves for pitching.

Moreover, as the ballast is increased, maximum GZ values decreased as shown in Figure 6. At lightship conditions in the heeling axis, the maximum GZ value was equal to 19.5 m and decreased to 2.2 m at 100% ballast. The maximum GZ value, when multiplied with the displacement of the structure can be used to find the maximum heeling moment

that the structure can sustain without capsizing. Beyond this point, the GZ values, which also correspond to the righting moment values of the structure, decreased significantly.

The decrease in GZ values could also be due to the decrease in metacentric height occurring with increasing ballast. The GZ curve was generated from the multiplication of the metacentric height with the angle of displacement. Therefore, as the position of the metacentric height, M, changed due to the decrease in WPA and the position of COG remained the same, the GZ also decreased.

The GZ curves obtained for the pitching axis can be seen in Figure 8. From 60% ballast, the GZ curve continuously flattened in shape until 100% ballast where the graph stopped abruptly at a pitch angle of 44.5°, unlike the rest of the ballast conditions. Under full ballast conditions the structure submerged totally at 44.5°.

The GZ curves for the pitching axis were made up of smoother curves. The lack of sharp peaks could be attributed to the symmetrical change in geometry during the submersion of the structure. First, the pontoons were submerged followed by the deck and the columns. In the heeling axis, the components on one side submerged first and the change in geometry as the structure submerged resulted in sharp peaks.

Similar to the heeling axis, the range of positive stability in the pitching axis increased steadily from 0% ballast to 50% ballast and started to decrease from 60% ballast onwards. However, the structure experienced a shorter range of positive stability in the pitching axis, with the maximum vanishing point occurring at a pitch angle of 57°, which is 10° lower than the value recorded for the heeling axis. In both the heeling and pitching axes, the structure experienced the largest range of stability at 50% ballast.

Moreover, it can be seen that the overall maximum GZ values of the pitching axis were lower than those of the heeling axis, resulting in a lower value of the maximum righting moment about the pitching axis. The lower value of the righting moment resulted in a smaller area present under the righting moment graph. This reduced the ability of the structure to withstand wind moments.

### 5.3. SA: RW

A rated wind speed of 12.5 m/s was utilised in the SA: RW analysis, corresponding to the scenario where the wind turbine exerts the highest axial thrust on the floater across its operating envelope. The values for the wind loads acting on the structure can be seen in Table 8.

**Table 8.** Wind loads acting on the structure.

| Parameter | Value |
|---|---|
| Rated wind speed | 12.5 m/s |
| Thrust force | $9.7 \times 10^5$ N |
| Moment due to thrust force | $1.2 \times 10^8$ Nm |
| Pitching Axis | |
| Wind loads acting on structure | $2.7 \times 10^5$ N |
| Moment due to wind loads acting on structure | $4.1 \times 10^6$ Nm |
| Heeling Axis | |
| Wind loads acting on structure | $4.3 \times 10^5$ N |
| Moment due to wind loads acting on structure | $1.3 \times 10^7$ Nm |

The wind moment about the heel axis, i.e., parallel to the rotor axis, was significantly smaller compared to the wind moment about the pitching axis, i.e., when the wind direction was perpendicular to the rotor. Such a difference occurred when the wind direction was parallel to the structure. The wind heeling moment accounts for only the wind loads on the structure and did not include the moment caused by the rotor thrust, which was much larger. The wind moment was calculated using Equation (3) and is denoted as a red line in Figures 8 and 9.

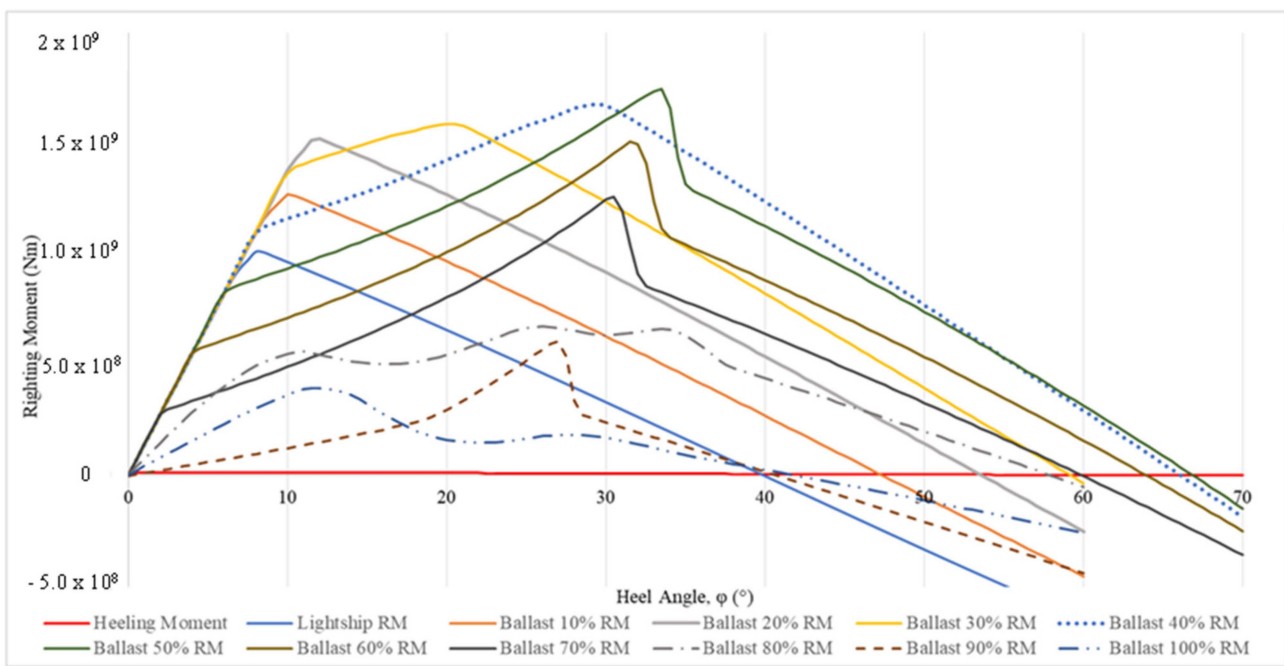

**Figure 9.** Moment curves about the heeling axis.

The DNV ST-0119 standard for floating wind turbines was adopted in conjunction with Equations (1)–(3). The area beneath the wind heeling and righting moment graphs was evaluated using numerical techniques.

The results of SA: RW with regards to the standard requirements are presented in Table 8.

The DNV ST-0119 standard requires that the following two conditions are met for a floating structure to be considered stable:

i.  Area under righting moment curve to the second intercept or downflooding angle in excess of the area under the wind heeling moment curve ≥ 130% (1.3).

ii. Righting moment curve should be positive over range of angles from upright to the second intercept.

Table 9 shows that the model passed all of the necessary standards requirements both in the heeling and pitching axes. The area under the righting moment curve was larger than the area under the wind moment curve in both axes. The values obtained for the pitching axis were significantly lower than those obtained for the heeling axis, a feature which could also be seen in the moment curves shown in Figures 9 and 10.

Furthermore, for all ballast conditions, the righting moment curve was positive until the second intercept, as was required by the standard.

The positive static stability values obtained from SA: RW showed that the structure conformed to all of the required DNV ST-0119 criteria and was stable from lightship up to 70% ballast capacity conditions. Passing the lightship condition implied that ballasting during the towing process was not necessary, thereby the lightship draught, being the smallest possible draught for the structure, made for an easier towing process due to lowest resistance. However, in the unlikely scenario where the structure may need to be ballasted inside the port, the tanks can be ballasted up to 60% capacity, at which point the draught is 8.2 m, while remaining within the port limits of length, breadth and draught (362 m, 62 m and 9.1 m). Therefore, the results of SA: RW further reinforced the notion that dual-pontoons were more prone to instability in the pitching axis. Moreover, the static stability results identified to which capacity it was possible to ballast the structure.

**Table 9.** Results of SA: RW.

| | | DNV ST-0119 Sect. 10.2.3 | |
|---|---|---|---|
| **Ballast** | **Result** | **Heeling Axis** | **Pitching Axis** |
| 0% | Pass | 48 > 1.3 | 5.6 > 1.3 |
| 10% | Pass | 87 > 1.3 | 6.0 > 1.3 |
| 20% | Pass | 135 > 1.3 | 6.3 > 1.3 |
| 30% | Pass | 221 > 1.3 | 6.3 > 1.3 |
| 40% | Pass | 279 > 1.3 | 6.4 > 1.3 |
| 50% | Pass | 300 > 1.3 | 6.4 > 1.3 |
| 60% | Pass | 271 > 1.3 | 5.7 > 1.3 |
| 70% | Pass | 899 > 1.3 | 4.5 > 1.3 |
| 80% | Pass | 274 > 1.3 | 8.4 > 1.3 |
| 90% | Pass | 451 > 1.3 | 3.7 > 1.3 |
| 100% | Pass | 136 > 1.3 | 2.9 > 1.3 |

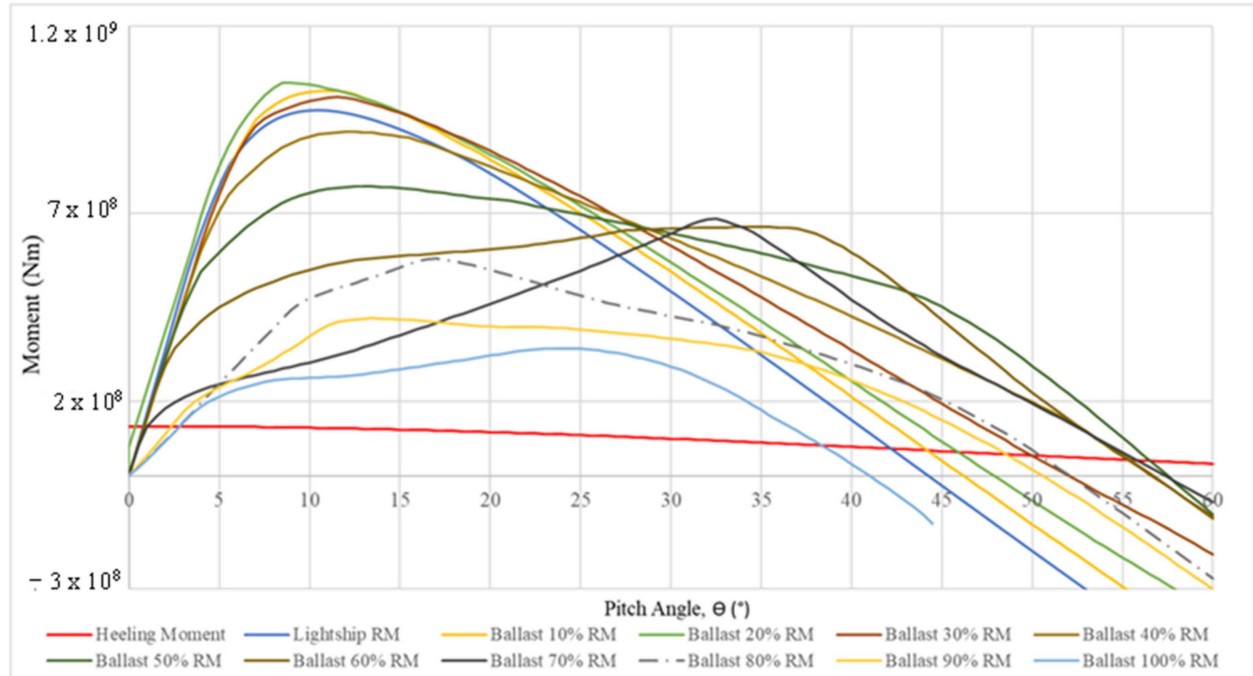

**Figure 10.** Moment curves about the pitching axis.

From SA: NW and SA: RW the following conclusions can be summarised:

- The structure showed positive results in both the heeling and pitching axes and passed all of the stability criteria set by DNV ST-0119.
- The results showed that the model presented better stability characteristics along the heel axis than along the pitch axis. This is common in dual-pontoon structures.
- The model experienced large pitch angles between ballast levels of 80–100%. This necessitated the dynamic balancing of the structure.

The large pitch angles present between 80% and 100% ballast were revealed to be the result of the shift in position of the longitudinal centre of floatation (LCF). When the pontoons were fully submerged, the WPA decreased sharply from 1538 m$^2$ at 70% ballast to 884 m$^2$ at 80% ballast, thereby causing the second order moment of the area of the waterplane to reduce significantly.

Although column-stabilised platforms are frequently used in offshore structures and are regarded as one of the safest floating platform types, the stability of such platforms is highly dependent on the WPA provided by the floater.

The WPA greatly affects the second order moment of area experienced by the structure, which is one of the key factors that affect the required righting moment for the structure to return to its original position. In the proposed design, the columns did not provide enough WPA to the structure and so the structure pitched forward until sufficient WPA was submerged again and a suitable value of the second order moment of area was regained for the structure to stabilise. Figure 11 presents a graphical representation of the effect of the decrease in WPA on parameters such as metacentric height, LCF and pitch angle.

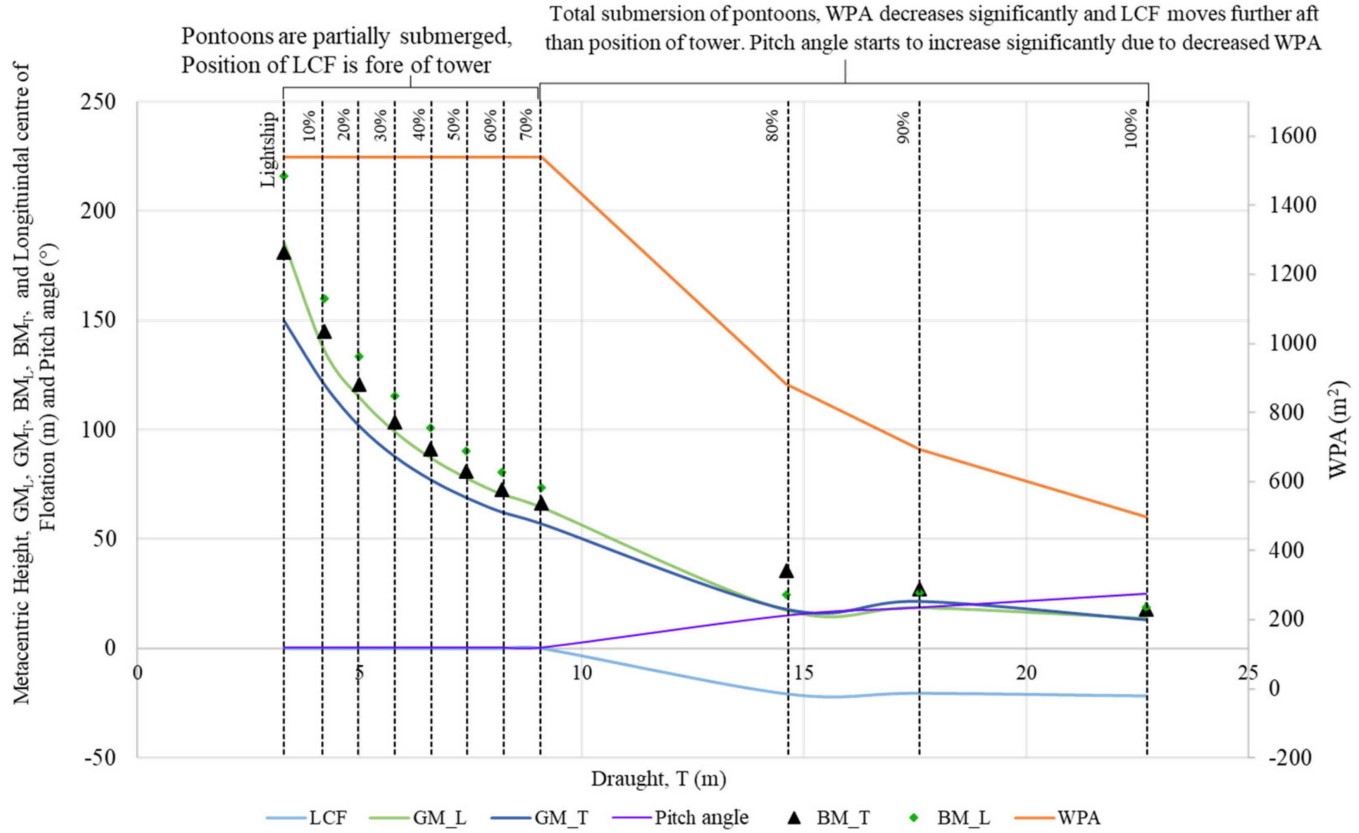

**Figure 11.** Effect of WPA on parameters.

The LCF is also sometimes referred to as the centroid of the WPA. Therefore, any changes in the WPA will also influence the LCF. The decrease in WPA caused the LCF of the structure to shift further aft, moving past the position of the tower and RNA as shown in Figure 12.

The decreasing WPA occurring at ballast conditions past 70% capacity could be said to have a significant negative effect on the overall stability of the structure, both in the pitching and the heeling axes. Increasing the ballast conditions past 70% capacity must be carried out accordingly in order to offset the large trim obtained. To prevent the excess pitch angle present between at 80% and 100% ballast, dynamic balancing is then required.

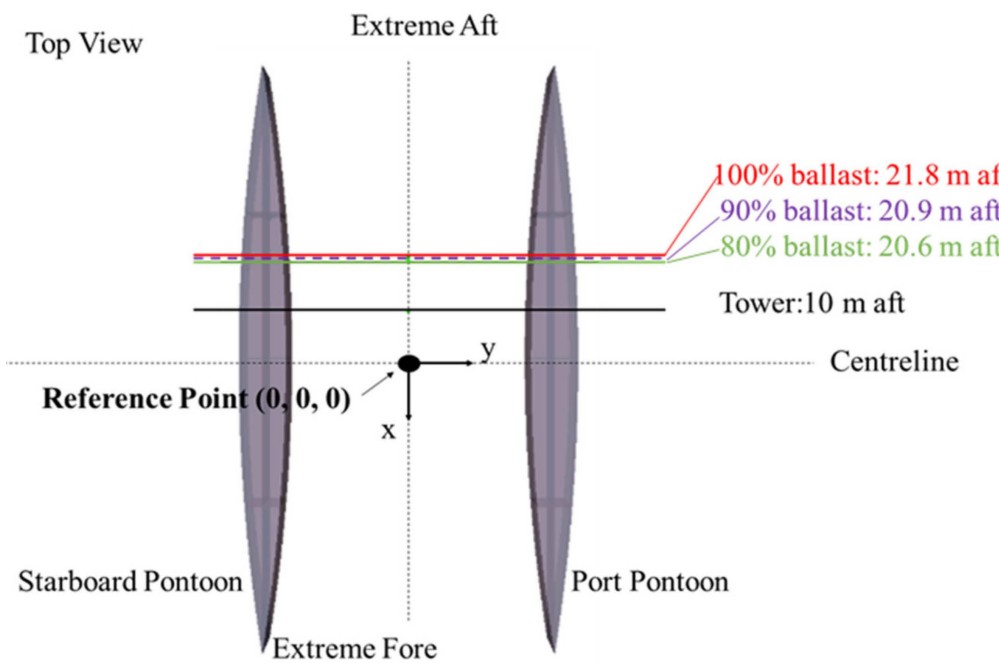

**Figure 12.** Shift in LCF.

*5.4. SA: PB*

From the results of SA: NW and SA: RW, it could be concluded that the proposed model could only be ballasted with equal tank capacity up to 70% ballast. Even though the structure passed all of the DNV ST-0119 requirements under all of the ballast states under no wind and rated wind speed conditions, between 80 and 100% ballast large pitch angles were present. Therefore, SA: PB was conducted to examine the possible ballast combinations to reduce the initial trim angle of the structure for the proposed design under rated wind speed conditions. The results are presented in Table 10.

**Table 10.** SA: PB results.

| Ballast Tanks | | Pitch, $\theta$ (°) | Heel, $\varphi$ (°) | $T$ (m) | $GZ_{max}$ (m) | Angle at $GZ_{max}$ (°) |
|---|---|---|---|---|---|---|
| Fore (F) | Aft (A) | | | | | |
| 80% | 80% | 15.2 | 0 | 14.6 | 3.8 | 17 |
| 80% | 85% | 14.6 | 0 | 14.8 | 3.9 | 17 |
| 80% | 90% | 13.8 | 0 | 14.9 | 4.0 | 17 |
| 80% | 95% | 12.8 | 0 | 15.0 | 4.0 | 18 |
| 80% | 100% | 9.6 | 0 | 14.7 | 4.3 | 20 |

Although ballasting the aft tanks at 100% capacity, the structure still experienced a pitch angle of 9.6°. Through partial ballasting, the LCF shifted forward as can be seen in Figure 13.

Furthermore, partial ballasting also increased the value of the maximum GZ and the angle at maximum GZ, resulting in a larger range of positive stability and an increase in the righting moment of the structure. As mentioned previously, with an increase in the maximum GZ value the righting moment of the structure also increases; therefore, the structure is able to withstand larger capsizing moments. Although the structure still experienced a large pitch angle, partial ballasting had an overall positive effect on the stability of the structure. The righting moment and the wind heeling moment of the partial ballast conditions analysed in SA: PB can be seen in Figure 14.

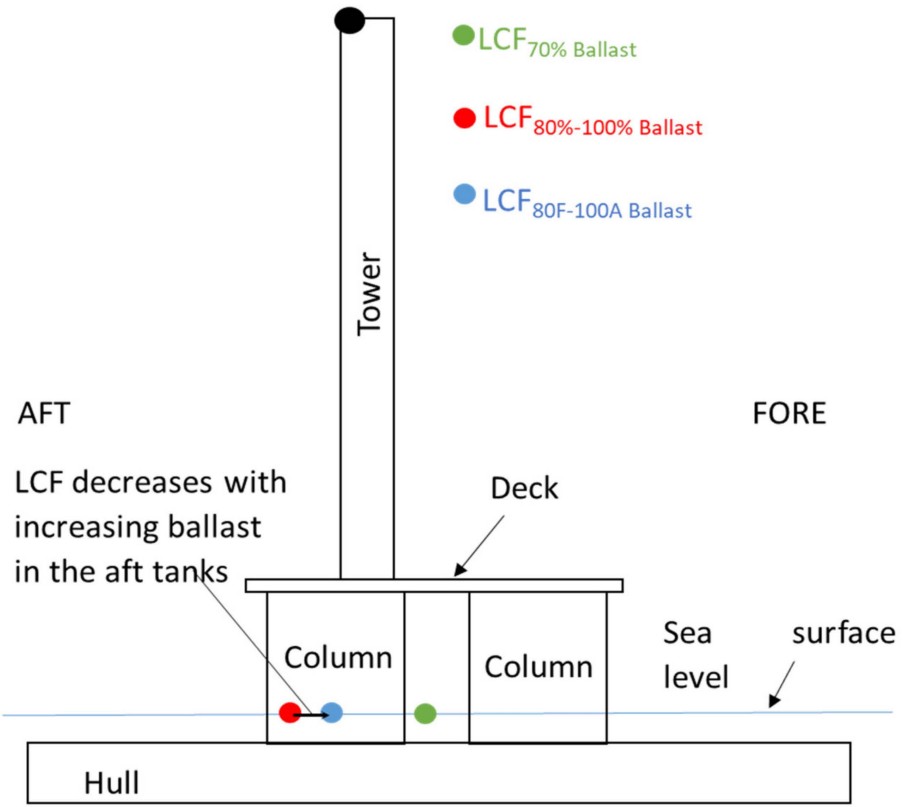

**Figure 13.** Shift in LCF position due to partial ballasting.

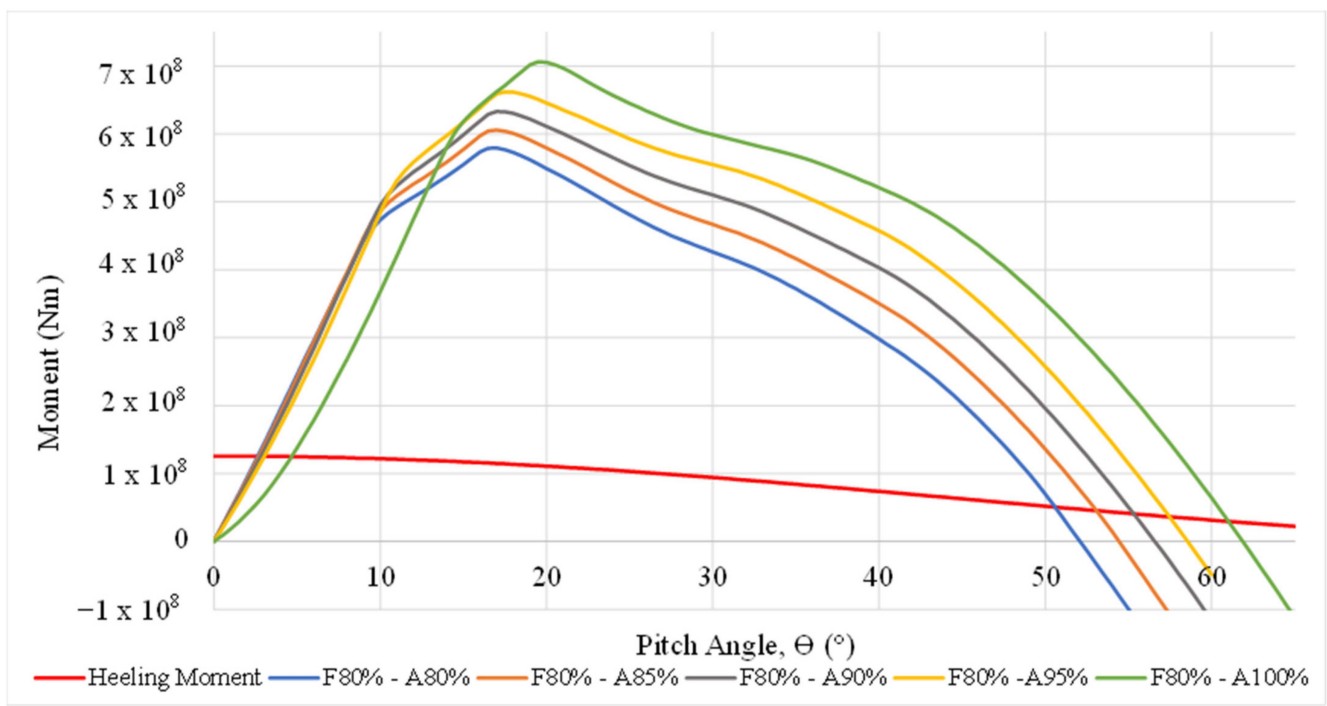

**Figure 14.** SA: PB—Righting moment and wind heeling moment curves.

The results of SA: PB with respect to the DNV ST-0119 standards can be found in Table 11.

**Table 11.** SA: PB—DNV ST-0119 standard check.

| Ballast | | Requirements | | |
|---|---|---|---|---|
| | | (i) | | (ii) |
| Fore | Aft | Pitching Axis | Result | Result |
| 80% | 80% | 3.7 > 1.3 | Pass | Pass |
| 80% | 85% | 4.0 > 1.3 | Pass | Pass |
| 80% | 90% | 4.3 > 1.3 | Pass | Pass |
| 80% | 95% | 4.6 > 1.3 | Pass | Pass |
| 80% | 100% | 4.9 > 1.3 | Pass | Pass |

## 6. Conclusions

The main aim of this paper was to analyse the hydrostatic stability of a novel dual-hull, self-aligning, semi-submersible structure with an SPM system. The proposed design passed all of the requirements set by DNV under no wind and rated wind conditions with different ballast conditions. The results and discussions presented in this paper can be summarised as follows:

- The theoretical values were successfully validated using the SESAM® software, hence confirming the correctness of the analysis.
- The structure experienced an initial forward pitch angle of 0.5° from 0% to 70% ballast capacity when in the equilibrium condition.
- The initial pitch angle increased significantly between 80% and 100% ballast, at which point the two pontoons were fully submerged and the WPA decreased significantly. Consequently, equal ballasting of the tanks was only possible up to a capacity of 70% to produce low acceptable pitching angles.
- Partial ballasting of the fore tanks at 80% ballast and the aft tanks at 100% ballast significantly reduced the large initial pitch angle present from 15.2° to 9.6°.
- All requirements stipulated by the DNV ST-0119 standard in the pitching axis when partially ballasted and under rated wind conditions were satisfied.
- The proposed concept exhibited greater stability characteristics in the heeling axis than in the pitching axis. This is synonymous with dual-pontoon structures.

The analysis also described the variation in important hydrostatic characteristics of GM and BM, in both the longitudinal and transverse conditions, LCF and WPA, for dual-pontoons floaters and the effect of small WPA at high ballast conditions and their influence on the effect of these variables on stability. Although the initial pitch angle was decreased and the structure passed all standard requirements set by DNV ST-0119, a large initial trim angle was still present.

DNV ST-0119 provides only a few criteria to be passed and does not provide a minimum value of GM in either rotation direction, or a maximum GZ value and the angle at which it occurs, or any value of area below the righting lever curves to specified angles of rotation as given for normal monohulled vessels. Therefore, it is left to the designer or class society to consider the importance of these values and establish the values that will ensure additional safety over and above that required by DNV ST-0119.

The proposed concept can be said to show promising results, despite the limitations present in this study, which include:

- The numerical study was restricted to hydrostatic stability under still water conditions and a rated windspeed.
- The proposed concept was not analysed under extreme wind conditions.
- Although the proposed concept is intended to be a self-aligning, SPM floating structure, the effects of mooring and the self-aligning characteristics of the structure were not taken into consideration.

Future work should be carried out to address the limitations present through:

- Further design iterations to reduce the excess initial pitch angle present at the higher ballast conditions.
- Dynamic analysis of the proposed concept under (1) combined extreme wind and wave actions and (2) conditions involving wind-wave misalignment
- Dynamic analysis for assessing the self-aligning capabilities under different metocean conditions
- Carrying out physical experiments using a scaled prototype in a wave tank facility to validate the results obtained from the numerical simulations.

**Author Contributions:** Conceptualization, D.S., C.D.M.M.-F. and T.S.; methodology, D.S., C.D.M.M.-F. and T.S.; software, D.S., C.D.M.M.-F. and T.S.; validation, D.S., C.D.M.M.-F. and T.S.; formal analysis, D.S., C.D.M.M.-F. and T.S.; investigation, D.S., C.D.M.M.-F. and T.S.; writing—original draft preparation, D.S.; writing—review and editing, D.S., C.D.M.M.-F., T.S., G.V., T.T.; visualization, C.D.M.M.-F. and T.S.; supervision, C.D.M.M.-F. and T.S.; project administration, C.D.M.M.-F.; funding acquisition, T.S. All authors have read and agreed to the published version of the manuscript.

**Funding:** The presented work has been supported through the Maritime Seed Award 2020 and the collaborative programme of the VENTuRE project. The latter project has received funding from the European Union's Horizon 2020 research and innovation programme (Project No. 856887).

**Institutional Review Board Statement:** Not applicable.

**Informed Consent Statement:** Not applicable.

**Data Availability Statement:** Not applicable.

**Conflicts of Interest:** The authors declare no conflict of interest. The funders had no role in the design of the study; in the collection, analyses, or interpretation of data; in the writing of the manuscript; or in the decision to publish the results.

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
