# Peer review of "Preliminary Analysis on the Hydrostatic Stability of a Self-Aligning Floating Offshore Wind Turbine"

_jmse, doi:10.3390/jmse10122017_

Round 1
Reviewer 1 Report
In this manuscript, the authors proposed a novel self-aligning foundation to support the large-capacity offshore wind turbine. As one of the key mechanism features, the hydrostatic performance was investigated via commercial software. Generally speaking, the study is reasonable, but the numerical algorithm seems to be a little ordinary but out of novelty. Therefore, this makes the manuscript more like a technical report. Besides, for the safety issue, is it better to examine the stability features under the survival cases with larger wind speeds?
Reviewer 2 Report
Review report of jmse-2055693
This paper analyzed the Hydrostatic Stability of a new Floating structure. No new method was developed in this paper. Still, it provides a complete analysis procedure and finds some useful conclusions that can support the design and operation of the new structure. However, the following comments should be considered:
Major comments:
## the main problem is that the authors should highlight the novel contributions of this paper. Please specify how the paper can contribute to the floating offshore wind field, and what are the new things conducted in this paper? It is very important, and please consider these comments carefully.
Specific comments:
#1. give the full name of SPM,
#2. Delete “available…sea.” In Abstract.
#3. Abstract: “favorable” is too subjective. Please avoid subjective descriptions.
#4. Introduction: “1.1 background” can be deleted.
#5. Introduction: Give references to the first two paragraphs. No references are listed for these two paragraphs…
#6. Introduction: “The main issue with upscaling…” give reference(s).
#7. Give the full name of “FOWT”.
#8. The upper line of Table 1 is missing.
#9. In lines 108-115, please provide a figure to illustrate MPM and SPM.
#10. “HyStOH is produced as a collaboration project…” in lines 131-132 can be deleted.
#11. One concern is how the sentences in lines 129-143 support the motivation of this paper? What are the motivations of the authors creating a new type of design, knowing that some designs (like X1) already exist? Please specify such things to highlight the novel contribution of this paper.
#12. The text in lines 175-178 seems not necessary.
#13. Full name of PSE
#14. Rearrange the conclusion part.
Round 2
Reviewer 2 Report
My comments have been addressed, the paper can be published now.
Author Response
The second revision was submitted
kind regards
Tonio
